# The Mediterranean Diet and the Western Diet in Adolescent Depression-Current Reports

**DOI:** 10.3390/nu14204390

**Published:** 2022-10-19

**Authors:** Magdalena Zielińska, Edyta Łuszczki, Izabela Michońska, Katarzyna Dereń

**Affiliations:** Institute of Health Sciences, College of Medical Sciences, University of Rzeszow, 35-959 Rzeszow, Poland

**Keywords:** adolescent depression, Mediterranean diet, Western diet, diet quality, adolescents

## Abstract

Depression is one of the most common mental disorders in the world and a current and growing social and health problem. The growing scale of the problem not only concerns adults, but now it particularly affects children and adolescents. Prevention, early diagnosis and treatment of mood disorders in adolescence is crucial because adolescent depression is a risk factor for recurrence of depression later in life, as well as many other mental health disorders in adulthood. The purpose of this study was to analyze data on the dietary patterns and composition of the Mediterranean diet as a modifiable risk factor for depression, which would be a viable prevention strategy and a good target for early intervention and supportive treatment of depression. Research shows that the Mediterranean diet pattern can reduce the risk and symptoms of depression, while western eating styles can increase the risk and severity of depression in adolescents. The number of studies in adolescent populations continues to increase, but most longitudinal and clinical studies are still insufficient. Modification of the diet can be a helpful strategy for the prevention and treatment of depression in adolescents; therefore, the diet of young people should be considered a key and modifiable goal in the prevention of mental disorders.

## 1. Introduction

Depression is one of the most common mental disorders in the world and is a current and growing social and health problem. The growing scale of the problem not only concerns adults, but now it particularly affects children and adolescents. Adolescence includes people aged 10 to 19 years [1]. This is a unique stage of human development and an important time to build the foundations for good health. Unfortunately, it can be disturbed and can be a predictor of psychopathology in adulthood, since most mental illnesses begin at the age of 14 and 1 in 7 young people aged 10–19 experience mental disorders. This is responsible for 13% of the global burden of disease in this age group [2]. Current data indicate that 1 in 4 young people worldwide experience clinically elevated symptoms of depression [3]. Comparing these results with pre-pandemic estimates suggests that the mental health problems of adolescents during the COVID-19 pandemic probably doubled. Furthermore, research results show that fears of depression and suicide have intensified during the pandemic, especially among teenage girls [4]. The two main diagnostic classification systems, the Diagnostic and Statistical Manual of Mental Disorders (DSM) and the International Statistical Classification of Diseases and Related Health Problems (ICD), are based on the identification of a number of key symptoms of depression. The diagnosis of depression in adolescents is made on the basis of the criteria described in the *Diagnostic and Statistical Manual of Mental Disorders*, Fifth Edition (DSM-5) [5]. The DSM-5 diagnoses major depressive disorder (the major form of depression) as a period of at least 2 weeks, during which there is a depressed mood or loss of interest or pleasure in almost all activities, plus at least four additional symptoms from a list that include weight changes, sleep disturbances, changes in psychomotor activity, fatigue, feelings of worthlessness or guilt, impaired concentration or decision-making ability, or thoughts of suicide. The clinical picture differs according to gender: adolescents report feelings of sadness, loneliness, irritability, pessimism, self-hatred, and eating disorders, while adolescents report somatic complaints, decreased ability to think or concentrate, lack of decision-making skills, anxiety and anhedonia [6]. Prevention, early diagnosis, and treatment of mood disorders in adolescence are of key importance because adolescent depression is a risk factor for the recurrence of depression later in life, as well as for many other mental health disorders in adulthood, such as anxiety disorders and bipolar disorders [7]. Presumed risk factors that can be modified in adolescence without professional intervention include the use of psychoactive substances, diet, and body weight [8,9]. The purpose of this study was to analyze data on the dietary patterns and composition of the Mediterranean diet as a modifiable risk factor for depression, which would be a viable prevention strategy and a good target for early intervention and supportive treatment of depression.

## 2. Materials and Methods

In this literature review, 119 scientific papers were analyzed. They were searched using databases: PubMed, Google Scholar, Researchgate and Science Direct. Research and review articles published from 2017 to the end of August 2022 were selected. Keywords used in the search are: “adolescent depression”, “mental health”, “adolescent diet”, “dietary habits”, “Mediterranean diet”, “Western diet”, “carbohydrate”, “protein”, “vitamin A”, “vitamin B”, “vitamin D”, “vitamin E”, “vitamin C”, “antioxidants”, “omega-3”, “iodine”, “iron” and “zinc”.

The inclusion criteria for the review were as follows:(1)Studies published in English.(2)Longitudinal, cross-sectional, case-control, clinical, interventional studies carried out mainly on humans, carried out in a group of adolescents.(3)Research on the relationship between diet quality or patterns and depression or major depressive disorder.

The exclusion criteria for review were:(1)Studies published in a language other than English.(2)Research carried out in groups other than adolescents.(3)Research on eating disorders and the relationship between body weight and depression.(4)Depression reported as a secondary problem to other diseases.(5)Studies that determined the effects of drugs or psychological treatments on depression.(6)Research on the influence of family practices and parental attitudes on food consumption and child mental health.(7)Studies with pregnant teenagers.(8)Animal studies.

## 3. Adolescent Depression as a Current Social and Health Problem

Adolescent depression can affect many areas of a teenager’s life, including social functioning, family relationships, and academic performance [10]. These problems can become chronic, leading to psychoactive and substance use disorders, accounting for approximately 40.5% of disability adjusted life years (DALY) in young people [11]. Furthermore, depression at this age is strongly associated with suicidal thoughts and attempts [12]. Suicide is the third leading cause of death in older adolescents (15–19 years), and more than 90% of teen suicides occur among young people living in low- and middle-income countries [13]. In 2019, approximately 1 in 6 young people reported having complied with a suicide plan last year, a 44% increase since 2009 [14]. This is a significant problem due to the impact of recent events and concerns about an increase in suicidal behavior [15]. The COVID-19 pandemic significantly increased the level of stress and depression in society [16,17,18,19,20,21,22,23,24]. The psychological impact of the COVID-19 pandemic has been widely discussed in recent months, and researchers have expressed concern about its potential negative consequences on mental health, especially among young people. Data to date show that the COVID-19 pandemic has affected the mental health of children and adolescents and is particularly associated with depression and anxiety in cohorts of young people [25,26,27]. The mean suicide mortality rates in Mediterranean countries are lower than the global average, which is why there is an interest in the influence of the Mediterranean diet on adolescent depression [13]. Its effectiveness in the context of prophylaxis, improving well-being, and support for the treatment of depression, has been studied primarily in the populations of adults [28,29,30,31,32,33] and the elderly [34,35,36,37]. Vegetative symptoms (changes in appetite and weight, loss of energy, and insomnia) are more common in depressed adolescents than in adults. Furthermore, there are significant etiological differences between adolescent and adult depressive disorders in terms of response to treatment. The evidence for the effectiveness of antidepressants in the treatment of adolescent depression is less than for depression in adults [38]. In particular, selective serotonin reuptake inhibitors (SSRIs) and tricyclic antidepressants show less potency in the treatment of depression in adolescents compared to adults [39]. The US Food and Drug Administration (FDA) has issued warnings for antidepressants in people up to 24 years of age due to the risk that these drugs may increase suicidal ideation and behavior [40]. Treatment is currently dominated by pharmacotherapy and psychotherapy (e.g., cognitive behavioral therapy, CBT), but such therapies are not entirely successful. A total of 50% of young people relapse or do not respond to therapy, suggesting that additional strategies are needed to prevent and treat mental disorders [41]. Special attention is needed to modifiable risk factors and the development of preventive strategies to reduce the overall burden of disease.

## 4. Mediterranean Diet-Characteristics and Potential in Adolescent Depression

Currently, the diet with the most evidence for a positive effect on depression is the Mediterranean diet (MD), which has recently been recognized as a promising treatment strategy to improve clinical outcomes in depression [42,43]. The Mediterranean diet pattern was identified in the 1950s and 1960s among European inhabitants of the Mediterranean basin. The populations of these countries were found to have reduced mortality and morbidity from various diseases [44]. MD is rich in plant foods such as vegetables, fruits, whole grains, nuts, seeds, and legumes, but also, to a lesser extent, in fish. Olive oil is the main source of dietary fat, while consumption of white meat, low-fat dairy products, eggs, and low-red wine at meals (in adults) is moderate and low consumption of red, processed meat, and sweets [44,45,46,47]. Important elements of this pattern of nutrition are also seasonality, biodiversity, and the use of traditional and local food products [47]. Contrary to the Mediterranean diet, the Western diet (WD)—currently dominant in developed countries—is based on convenience and highly processed foods and is characterized by high consumption of processed and red meat, desserts and sweets, fried foods, high-fat dairy products, refined grains and low consumption of vegetables, fruits, whole grains, and legumes [47]. Therefore, the Western diet is the reverse and opposite of the principles of the Mediterranean diet. In pyramid form, this relationship is shown in Figure 1.

Recent meta-analyses of studies provide convincing evidence that people with an inflammatory eating pattern are at increased risk of developing depression over time [42,48]. Furthermore, according to a recent meta-analysis by Orlando et al., a healthy eating pattern was significantly associated with a lower number of depressive symptoms in children and adolescents [49]. Many studies have shown that young people’s diets are geared toward WD. Adolescents eat foods with poor nutritional value, foods high in sugar and saturated fatty acids (SFA) [50]. Modification of pro-inflammatory diets typically associated with mental disorders to a more Mediterranean anti-inflammatory diet regimen in adolescents could be a new strategy to counter the inflammation associated with the onset and severity of mental disorders. Recent analyses extend this aspect and describe the influence of obesity mechanisms and entero-microbial interactions on the development and progression of mood disorders [51,52]. Furthermore, in a study by Dehghan et al. a significant inverse relationship was found between the dietary antioxidant index (DAI) and depression in adolescent girls, which underscores the importance of a healthy and anti-inflammatory diet for the mental health of adolescents [53]. Research determining the relationship between eating and mood disorders can be divided into 2 categories: analysis of eating behavior/patterns and analysis of nutrient intake/condition.

## 5. Analysis of Eating Behavior/Patterns in the Context of the Mediterranean Diet

The noticeable eating patterns (poor appetite, skip meals, and predominant craving for sweets) that precede depression are the same as those that occur during depression [51]. Korczak et al., in a recent cross-sectional study, confirmed that nutritional practices among children and adolescents with major depressive disorder are different from those of other mental health conditions and healthy adolescents. These adolescents consume less healthy food than their nondepressed peers and may have poorer quality diets compared to adolescents with mental disorders related to the absence of milk [54]. Khalid et al. described the relationship between diet quality and mental health disorders among children and adolescents and showed a correlation between unhealthy, low-quality diets and depression or poor mental health [55]. A study by Voltas et al. showed higher symptoms of depression in girls whose diet was the least similar to the Mediterranean diet [56]. The Australian Childhood Determinants of Adult Health (CDAH) cohort study found the association between the diet of adolescents aged 10 to 15 years and mood disorders over a 25-year follow-up period. These results do not support the hypothesis that a higher quality of the diet at an early age is associated with a reduced risk of mood disorders in adulthood [57]. Winpenny et al. showed that the quality of the diet of adolescents was not associated with depressive symptoms after adjusting for covariates [58]. Despite different reports, it is worth paying attention to the growing amount of current data confirming the impact of diet quality on mental health [59]. Pro-inflammatory eating patterns are associated with an increased risk of mental health problems, including depressive symptoms in adolescents, through biologically plausible pathways of obesity (adipose tissue releases inflammatory cytokines) and inflammation [48,60]. In a study by Oddy et al., Western diets (high consumption of red meat, takeaways, refined foods and sweets) at 14 years of age were associated with higher energy and BMI intakes, and therefore with BMI and biomarkers of inflammation at 17 years of age (*p* < 0.05), and therefore with depressive symptoms [60]. Increased levels of pro-inflammatory cytokines are observed in children and adolescents with major depressive disorders [61,62,63]. Increased pro-inflammatory markers have been found to predict depression symptoms in children and adolescents, and unhealthy eating behavior can exacerbate depression symptoms by increasing inflammation [64]. A reduction in systemic inflammation due to the Mediterranean diet has been observed in adolescents [65]. The influence of the Mediterranean diet on chronic diseases may be related to a reduction in inflammation, measured mainly by CRP and IL-6 markers in epidemiological cohorts of adolescents [66]. Furthermore, in a recent study by Sured et al., the authors observed that among adolescents, greater adherence to the Mediterranean diet was associated with lower CRP levels [48]. It is worth highlighting that more research is needed to clarify whether the same inflammation marker is associated with the Mediterranean diet in adolescents. Increased adherence to the Mediterranean diet can counteract the effects of stress-induced inflammation and reduce the risk of future mental health [67]. Research confirms that in children and adolescents, greater adherence to MD is associated with a better academic performance, while WD at 14 is associated with poorer cognitive outcomes at 17 years of age [68]. A Western style of eating characterized by high consumption of fries, hydrogenated fats, mayonnaise, sweets, desserts, high-fat dairy products, refined grains, red or processed meats, pickles, offal and soft drinks, and low consumption of low-fat dairy products was positively correlated with depression scores among adolescents [69]. There are several reports showing that neurochemistry and brain function may be influenced by saturated fatty acids (SFAs). It should be noted that SFAs have been shown to disrupt many brain circuits related to mood regulation, such as inflammation of the nervous system [70]. Diet can exacerbate and alleviate oxidative stress by depriving or increasing the supply of compounds with antioxidant properties. The health-promoting properties of MD are related to the content of polyphenols in it. Polyphenols have been shown to modulate the gastrointestinal microflora and act as anti-inflammatory compounds [71]. The effects of individual nutrients (e.g., fiber, polyunsaturated fatty acids, and polyphenols) on brain health may also depend on their direct effect on microflora [72,73]. Research shows that a Mediterranean diet with a large supply of dietary fiber influences the diversified intestinal microflora and is associated with a reduced likelihood of depression [74]. Interestingly, MD and other diets rich in fruits and vegetables have also been shown to have a beneficial effect on the intestinal microbiota [75]. They aim to reduce the inflammatory potential of the diet and are also associated with a reduced risk of depression [76,77,78,79,80]. In addition, research suggests that there is a dose–response relationship by which any increase in fruits and vegetables is associated with improved mental health. Iranian adolescents who followed a pro-inflammatory diet were more likely to develop depressive symptoms [80]. In a narrative review by Chopra et al., Mediterranean and traditional diets rich in complex carbohydrates and omega-3 fatty acids showed a negative correlation with the appearance of depression. On the other hand, western eating patterns (sweetened beverages, processed / unhealthy foods, high in saturated fatty acids and trans fatty acids) along with low fruit and vegetable consumption have been associated with an increased risk of depression among adolescents [59]. Tehrani et al. showed that the adherence to Mediterranean eating patterns was associated with a lower risk of depression and a reduction in depressive symptoms in female adolescents [81]. Furthermore, current scientific reports show that a higher dietary phytochemical index (DPI) score is associated with a lower risk of depression [82]. Research shows that the Mediterranean diet pattern can reduce the risk and symptoms of depression, while Western eating styles can increase the risk and severity of depression in adolescents. Table 1 presents the current results of studies among adolescents on eating behavior and its impact on depression.

## 6. Analysis of Nutrient Intake/Condition in the Context of the Mediterranean Diet

Epidemiological evidence suggests an association between depression, as well as other mental illnesses, and nutrient deficiencies [93,94]. Although the specific mechanisms of the role of the Mediterranean diet in the prevention of depression are not fully understood, several hypotheses have been described that may justify this relationship. One of them describes that following the Mediterranean diet guarantees an adequate supply of nutrients [80]. Although all nutrients are essential for the human brain, key ingredients support the development of the nervous system. They include protein, iron, choline, folic acid, iodine, vitamins A, D, B6, and B12, and long-chain polyunsaturated fatty acids [94]. The main sources of protein in the Mediterranean diet are fish, seafood, and legumes [95]. Dietary protein intake (individual amino acid intake) can affect brain function and mental health. In addition, many of the neurotransmitters in the brain are made of amino acids. For example, the amino acids tryptophan and tyrosine are precursors of the neurotransmitters serotonin and dopamine [96]. Table 2 shows the current available clinical trials on macronutrient intake in the diet of adolescents with respect to depression.

Polyunsaturated omega-3 fatty acids may be a potential aid in the treatment of depression, as n-3 PUFAs have been shown to reduce inflammation in the nervous system through their antioxidant and anti-inflammatory properties [97]. N-3 PUFAs are the most important bioactive nutrients in fish, nuts, and olive oil, which are essential components of the MD; however, clinical trials have focused on supplementation with these acids, not just a proper nutritional supply [94]. Numerous studies have shown that n-3 PUFA relieves and prevents depression symptoms [59,98]. Table 3 presents current clinical studies on polyunsaturated fatty acids in relation to juvenile depression.

Vegetables and fruits are the staple of the Mediterranean diet and contain many nutrients, including vitamins A, C, K, E, B6, folic acid, copper, magnesium, iron, thiamine, niacin, and choline. Another element of the MD are whole grains, fish and seafood, which are also a source of zinc, iodine, magnesium, copper, and many other nutrients [95]. Oxidative stress plays an important role in depression. Antioxidants inhibit the lipid peroxidation process by inactivating free radicals. Low levels of antioxidants, mainly vitamins A, E and C, appear to be a possible cause of depressive disorders [105]. The risk of depression can be exacerbated by low serum vitamin D levels. Research indicates that vitamin D also acts as a neuroactive steroid that plays a key role in neurotransmitter expression, regulation of neurotrophic factors, neuroimmunomodulation, and the production of neurotrophic factors, which makes it biologically probable that vitamin D may be strongly associated with depression [106]. Homocysteine, an amino acid derived from ingested methionine, is a component of the homocysteine-methionine cycle that mediates methylation and plays a key role in maintaining the biochemical balance within the central nervous system. Hyperhomocysteinemia affects neurological functions and occurs in people with depression [107]. Zinc is a micronutrient that has received a lot of attention due to its possible role in depression: for example, zinc dysregulation in the hippocampus, amygdala, and cortex is allegedly associated with the pathophysiology of depression [108]. Table 4 shows the current available clinical studies on vitamins and minerals in relation to juvenile depression.

Hoepner et al. describe a number of additional supplements, i.e., S-adenosylmethionine (SAMe), L-acetylcarnitine, alpha-lipoic acid, N-acetylcysteine, L-tryptophan, coenzyme Q10, inositol, which may play a role in the fight against depression in patients with an insufficient response to therapy antidepressants [116]. According to the author of the study, supporting pharmacotherapy in depression with supplements aimed at nutritional and physiological factors can intensify the antidepressant effect. Among adolescents, there are reports of calcium supplementation with L-methylfolate and N-acetylcysteine. Rainka et al. showed that calcium L-methylfolate can provide therapeutic benefits in many mental diseases, including depression, and is well tolerated in the adolescent population [117]. Nery et al. describe that N-acetylcysteine significantly improved the results of depression and anxiety symptoms and the overall clinical impression in adolescents (all *p* < 0.001), but the study was carried out in a small group (9 participants) [118]. Cullen et al. described similar results, showing that the results of depression and psychopathology (but not the results of impulsivity) decreased after treatment with N-acetylcysteine [119]. Furthermore, the author describes that N-acetylcysteine may be a promising treatment option for adolescents with non-killing self-harm (NSSI). Furthermore, a new area of research on depression is the use of probiotics [120,121,122,123]. The described data are promising directions for future research and show how strongly the field of nutritional psychiatry is developing [41].

## 7. Limitations and Future Directions

The review notes the predominance of cross-sectional studies, clearly indicating the need for further longitudinal and clinical studies in adolescents in relation to the quality of their diet and its impact on adolescent depression. In addition, the review presented focuses on diet quality in relation to prevention, early intervention and supportive treatment of depression. As can be seen from the data presented, the impact of diet on the prevention of adolescent depression and its importance in early intervention is highlighted. A valuable direction for further research would be to consider body dissatisfaction, eating disorders and adolescent weight in relation to the quality of their diet, which were exclusion criteria for this review. In addition, it would be useful to take into account energy and nutritional deficiencies, which may also contribute to the risk of the onset and course of depression. In further studies, it will be worth considering health promotion interventions and programs to improve young people’s awareness of the impact and importance of diet on mental health on eating disorder behavior.

## 8. Conclusions

Taking into account the topicality and growing prevalence of adolescent depression and the unhealthy lifestyle of adolescents, it is important to identify appropriate nutritional strategies to prevent and support the treatment of depression. The most current research shows an association between diet quality and eating patterns and the risk of adolescent depression. The Western diet is associated with an increased risk of depressive symptoms in adolescents, while following the Mediterranean diet can reduce depressive symptoms and improve rates or remission. However, more research is needed to examine the relationship between diet and depression among young people to further elucidate the role of diet in the development and symptoms of depression. The number of studies in adolescents continues to increase, but most longitudinal and clinical studies are still insufficient. There is a lack of research in the depressed adolescent population on the level and consumption of nutrients such as magnesium, iodine, and selenium. Modification of the diet can be a helpful strategy for the prevention and treatment of depression in adolescents; therefore, the diet of young people should be considered a key and modifiable goal in the prevention of mental disorders.

## Figures and Tables

**Figure 1 nutrients-14-04390-f001:**
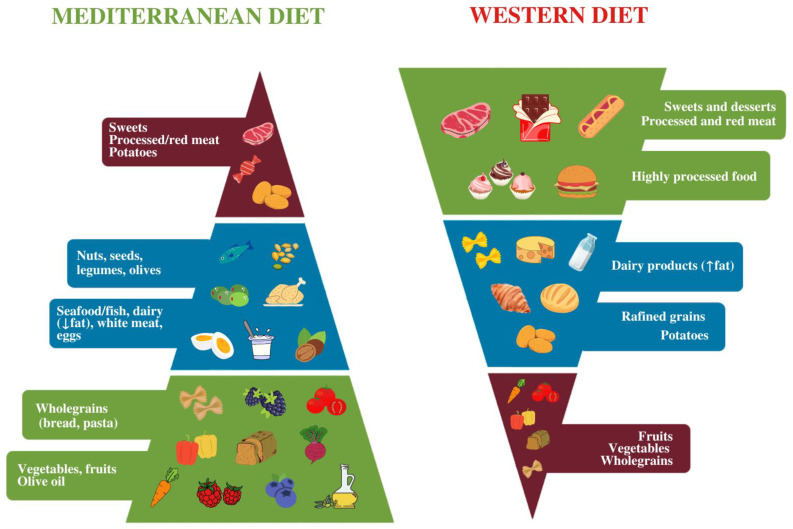
Characteristics of the Mediterranean and Western diet.

**Table 1 nutrients-14-04390-t001:** Dietary behaviors in the context of the Mediterranean diet and their influence on depression in a study among adolescents.

Authors	Year of Publication	Part of the Mediterranean Diet	Age and Size of the Study Group	Results and Conclusions
Hong et al. [83]	2017	Vegetables, fruits	*n =* 65,212 Age: 12–18 years (mean age = 15.1 years, SD = 0.02)	Healthy eating behavior (regular consumption of fruit, vegetables, breakfast and milk) was negatively associated with perceived symptoms of stress and depression.
Winpenny et al. [58]	2018	Vegetables, fruits, and fish	*n =* 603 Age: 14 years (study enrollment) and 17 years (follow-up)	There were no significant associations between diet quality (MD), fruit and vegetable or fish consumption, and depression symptoms at 14 and depressive symptoms at 17, corrected for baseline depression symptoms.
Oddy et al. [60]	2018	Fruits, vegetables, fish, whole grains	*n =* 843 Age: 14 years (study enrollment) and 17 years (follow-up)	A diet rich in fruits, vegetables, fish and whole grains was inversely correlated with BMI and inflammation at 17 years of age (*p* < 0.05). The Western nutritional pattern is indirectly associated with an increased risk of developing depression symptoms in adolescents through the biological pathways of obesity. and inflammation, and a “healthy” diet appears to be protective in these pathways.
Hoare et al. [84]	2018	Vegetables, fruits	*n =* 3696 Age: 16 years (study enrollment) and 29 years (follow-up)	Fruit consumption was cross-sectionally associated with a lower probability of developing depression in adolescence in both men and women, both before and after controlling the covariates. Vegetable consumption among women was cross-sectionally associated with a decreased probability of developing depression in adolescence. Those who never experienced depression were the largest consumers of fruits and vegetables during adolescence.
Ferrer-Cascales et al. [85]	2018	Breakfast, cereal products, dairy products	*n =* 527 Age: 12–17 years (mean age = 14.30 years, SD = 1.52)	A high-quality breakfast, characterized by the consumption of grain products and dairy products, was associated with lower levels of perceived stress and depressive symptoms in adolescents. People who skipped breakfast showed lower levels of stress and depression than breakfast eaters who ate a low-quality or very low-quality breakfast.
Zhu et al. [86]	2019	Breakfast	*n =* 10 174 Mean age: 19.76 years, SD = 0.86	Skipping breakfast was associated with an increased risk of depressive symptoms. and these relationships did not change after adjusting for many potentially confounding variables. Eating breakfast is crucial to reducing the incidence of depressive symptoms.
Khayyatzadeh et al. [87]	2019	Vegetables, fruits, fish, dairy products	*n =* 670 teenage girls Age: 12–18 years (mean age = 14.5 years, SD = 1.5)	The high consumption of fruits, vegetables, fish, and dairy products was associated with a lower incidence of depressive symptoms. However, no significant links have been found between traditional and Western eating patterns as a result of depression.
Tanaka et al. [88]	2019	Vegetables	*n =* 858 Mean age: 15.49 years, SD = 1.78)	Regular consumption of green and yellow vegetables reduces the symptoms of depression in adolescents. Eating vegetables in these colors can be crucial for the mental health of adolescents.
Khayyatzadeh et al. [89]	2021	Fiber	*n =* 988 teenage girls Age: 12–18 years (mean age: 14.5 years, SD = 1.52 and 1.54)	The consumption of soluble and insoluble dietary fiber was much higher in healthy adolescents compared to people with symptoms of depression (*p* < 0.001). There was a significant inverse relationship between dietary antioxidant intake and depression symptoms among Iranian teenagers.
Gao et al. [90]	2021	Breakfast	*n =* 1017 Mean age: 19 years, SD = 18.19	Teenagers who skipped breakfast more than once a week had an increased risk of developing depressive symptoms compared to those who ate breakfast every day. Eating snacks between meals, desserts, and sugary beverages was significantly related to depression in univariate analyzes.
Cao et al. [91]	2022	Breakfast	*n =* 3967 Age: 11–19	Lifestyle is associated with the risk of developing depressive symptoms, including skipping breakfast in both girls and boys.
Sangouni et al. [92]	2022	Breakfast	*n =* 933 teenage girls Age: 12–18 years	There was a significant difference between the depression score categories for main meal consumption (*p* < 0.001) and regular meal consumption (*p* < 0.001). There was no significant relationship between breakfast consumption and the depression score (*p* = 0.007), snack consumption (*p* = 0.002), and consumption of fried foods (*p* > 0.05).

**Table 2 nutrients-14-04390-t002:** Macronutrients in the diet of children and adolescents suffering from adolescent depression.

Authors	Year of Publication	Age and Size of the Study Group	Results and Conclusions
Macronutrients:			
Khanna et al. [95]	2020	*n =* 546 Age: 13–15 years, SD = 0.5	Low consumption of protein-rich foods such as milk and legumes was significantly associated with higher mean depression scores. Stronger relationships have been observed with milk protein consumption, especially during breakfast meals.
Farhadnejad et al. [96]	2020	*n =* 263 teenage girls Age: 15–18 years (mean age: 16.20 years, SD = 0.97)	Participants with depression had lower protein intakes (% energy) and a higher proportion of carbohydrates in the diet (*p* < 0.05) compared to people without depression.

**Table 3 nutrients-14-04390-t003:** Polyunsaturated fatty acids in the diet of children and adolescents in adolescent depression.

Authors	Year of Publication	Age and Size of the Study Group	Results and Conclusions
Fatty Acids:			
Trebatická et al. [99]	2017	*n =* 35 Age: 11–17 (mean age: 15.5 years, SD = 1.5) *n =* 17 omega-3 group, *n =* 18 omega-6 group)	Significant decreases in depression scores were observed in the 35 patients who completed the 12-week intervention after the 12-week intervention only in the Omega-3 group (*p* = 0.034). Fish oil (57.2% EPA and 42.8% DHA) may be an effective supplement to standard antidepressant therapy. in the treatment of depressive disorders in adolescents.
Gabbay et al. [100]	2018	*n =* 48 Age: 12–19 years (*n =* 21—omega-3 group, *n =* 27—placebo group)	A 10-week randomized, placebo-controlled study of omega-3 fatty acids in adolescents with severe depression (MDD) did not show improvement in response to omega-3 fatty acids (final dose = 3.6 g/day, EPA to DHA ratio 2:1) compared to placebo for the severity of depression, anhedonia, irritability, and suicidal tendency.
van Wurff et al. [101]	2020	*n =* 257 Age: 13–15 years (mean age: 14.11 years, SD = 0.55)	No evidence was found to link DHA, EPA and the omega-3 index (O3I) to depression in adolescents. One possible explanation for the lack of association in the present study could be that the design of this study was participant selection. with O3I ≤ 5%.
Katrenčíková et al. [102]	2020	*n =* 78 (58 depressed patients aged 15.6 ± 1.6 years, 20 healthy people aged 14.8 ± 2.4 years)	Supplementation with omega-3 fatty acids increased non-atherogenic HDL subfractions. HDL-CH and its subfractions, but not LDL-CH, may play a role in the pathophysiology of depressive disorders.
Trebatická et al. [103]	2020	*n =* 58 Age: 11–17 (mean age: 15.7 years, SD = 1.6) (*n =* 29 omega-6, *n =* 29 omega-3)	An emulsion of fish oil rich in omega-3 acids can be an effective supplement to standard antidepressant therapy in the treatment of depressive disorders in adolescents.
Paduchová et al. [104]	2021	*n =* 58 Mean age: 15.6 years, SD = 1.6	There was a significant positive correlation between the severity of depression or the omega-6/omega-3 FA ratio and plasma thromboxane B and a negative correlation with brain-derived neurotrophic factor (BDNF). Children and adolescents with depressive disorders had higher levels of thromboxane B and decreased vitamin D levels compared to healthy controls. Supplementation with omega-3 FA in conjunction with standard antidepressant therapy may have a beneficial effect on thromboxane levels. However, the positive effect of omega-3 FA supplementation on BDNF levels was observed only in patients with depression.

**Table 4 nutrients-14-04390-t004:** Vitamins and minerals in the diet of children and adolescents with adolescent depression.

Authors	Year of Publication	Age and Size of the Study Group	Results and Conclusions
Antioxidant Vitamins (A, C, E):			
Bahrami et al. [109]	2019	*n =* 563 teenage girls Age: 12–18 years (mean age 14.5 years, SD = 1.5)	There were no differences between people with high and low depression scores with respect to vitamins A, E, and their corrected lipid levels (*p* > 0.05).
Farhadnejad et al. [96]	2020	*n =* 263 teenage girls Age: 15–18 years (mean age 16.20 years, SD = 0.97)	Higher β-carotene consumption of -carotene was associated with a lower incidence of depression, anxiety, and stress. Vitamin C consumption was not associated with the risk of depression and anxiety stress and stress. A higher vitamin E intake was associated with a lower risk of stress.
Khayyatzadeh et al. [89]	2021	*n =* 988 teenage girls Age: 12–18 years (mean age: 14.5 years, SD = 1.52)	People with no or minimal depression symptoms had a significantly higher intake of α-carotene (*p* = 0.01), β-carotene (*p* = 0.006), lutein (*p* = 0.03) and vitamin C (*p* = 0.04) in compared to people with mild to severe symptoms of depression. Higher dietary intakes of vitamin C, but not vitamin E, were associated with fewer symptoms of depression among adolescents.
**Vitamin D:**			
Bahrami et al. [110]	2018	*n =* 988 Mean age: 14.56 years, SD = 1.53	Vitamin D capsules (50,000 IU of D3/1 time per week for 9 weeks) showed a significant reduction in the total depression score of BDII (*p* = 0.001). A high dose of vitamin D once a week can be useful in alleviating symptoms of depression.
Libuda et al. [111]	2020	*n =* 113 Age: 11.0–18.9 years (*n =* 56 vitamin D supplementation, *n =* 57 placebo group)	Vitamin D supplementation (2640 IU of vitamin D once a day) in children and adolescents who suffered from depression with vitamin D deficiency did not result in a significant reduction in self-reported depression symptoms, but a significant reduction in parent-reported depressive symptoms after 4 weeks of stationary or daily treatment compared to placebo.
Esnafoglu et al. [112]	2020	*n =* 89 with a mean age of 15.08, SD = 1.46 with depressive disorders, *n =* 43, mean age control group 14.41 years, SD = 2.32	Vitamin D levels were significantly lower in the group of teens with depressive disorders (*p* < 0.001). Lower vitamin D levels have been shown to play a role in the pathogenesis of depression in adolescents. It is recommended that clinicians test vitamin D levels in adolescents with depression.
Al-Sabah et al. [113]	2022	*n =* 704 Age: 11–16 years Mean age: 12.25 years, SD = 0.8	There was no significant correlation between serum 25(OH)D concentration and the result of depression symptoms measured with the Pediatric Depression Inventory (CDI).
**Vitamins of group B:**			
Esnafoglu et al. [112]	2020	*n =* 89 with a mean age of 15.08, SD = 1.46 with depressive disorders, *n =* 43, mean age control group 14.41 years, SD = 2.32	There were no significant differences between the groups in terms of folate levels (*p* = 0.052). The depressed group had extremely low vitamin B12 levels compared to the control group (*p* < 0.001). Low vitamin B12 levels and elevated homocysteine levels may play a role in the pathogenesis of depression in adolescents.
**Minerals:**			
**Zinc:**			
Tahmasebi et al. [114]	2017	*n =* 100 Age: 15–20 years (mean age: 17.9 years, SD = 1.2)	Each 10 μg/dL increase in serum zinc resulted in a 0.3 and 0.01 decrease in depression and anxiety scores, respectively (*p* < 0.05). Serum zinc levels were inversely correlated with mood disorders, including depression and anxiety in adolescents.
Gonoodi et al. [115]	2018	*n =* 408 Age: 12–18 years (mean age: 15.2 years, SD = 1.5)	Dietary zinc consumption (7.04 ± 4.28 mg/day) was significantly lower among those with mild or severe symptoms of depression than among those with no or minimal symptoms of depression (8.06 ± 3.03 mg/day). Dietary zinc consumption was inversely correlated with the depression score (r = 0.133, *p* = 0.008). Dietary zinc consumption, but not serum zinc concentration, was inversely related to depressive symptoms.

## Data Availability

Not applicable.

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
