# Peer review of "The Mediterranean Diet and the Western Diet in Adolescent Depression-Current Reports"

_nutrients, 2022, doi:10.3390/nu14204390_

Round 1
Reviewer 1 Report
I read your paper with interest, I think it has a good quality of presentation, proper English translation and significance of content, and interesting point of view for researchers.
I recommend the following be corrected:
- figure 1 has to be redone, the 2 types of diets seem similar.
- write down how many articles were read in total.
- add some biases you think the study has.
The overall merit and interest in this paper are good.
I accept it after minor revision.
Author Response
Thank you to the esteemed Reviewer for your valuable guidance. The answers have been sent in the attachment.

Reviewer 2 Report
This is a potentially useful review of literature bearing on the putative benefits of the Mediterranean diet in prevention, early intervention and treatment programs for depression in adolescents. While the idea that diet might influence mental health is not new (e.g., https://www.biblio.com/book/psychodietetics-food-key-emotional-health-cheraskin/d/780516888), I was interested to see whether and how research in this field had progressed. I have just a few comments that can be taken as suggestions for how the manuscript might be improved as part of a revision, should this be the Editorial decision:
1) Clarification of the aim of the research would be helpful. In the Abstract it is stated that “The purpose of the study was to analyze current data on dietary patterns and the composition of the Mediterranean diet for adolescent depression” but noting the aim of the study at the end of the Abstract appears odd. There needs to be a clear statement of the study aim at the end of the Introduction section. Further, the statement that is currently at the end of the Introduction (“Diet is a modifiable risk factor for depression … research interest”) appears to put the cart before the horse, i.e., wasn’t the aim of the study to review evidence in this regard? It would also be helpful to clarify the study methodology in both the Abstract and the Introduction. From the Materials and Methods section it appears that the authors conducted a review of relevant literature but nowhere is it made clear what sort of review was conducted, e.g., systematic, scoping, narrative, etc. I gather that the authors are interesting in the potential role of diet in not only prevention programs but also early intervention and treatment programs. If so, this should also be made clear.
2) A major concern, and significant limitation of the research, for me is the fact that “Research on eating disorders and the relationship between body weight and depression” is among the exclusion criteria for the research reviewed. Given the well-known associations between body dissatisfaction, eating-disordered behavior, body weight and mental health in adolescents (see https://pubmed.ncbi.nlm.nih.gov/28389062/;
https://pubmed.ncbi.nlm.nih.gov/26880693/), I cannot see how exclusion of such research makes sense when considering potential associations between diet and mental health. Rather, it seems that factors playing a key role in … are being ignored. Ideally, the search would be repeated with no such exclusion criterion and the text of all sections of the manuscript revised accordingly. At the very least, however, and if the authors prefer to retain their exclusion criteria as is, then it should be made clear in the Discussion and/or Conclusion sections that this is a limitation of the research and why. If this sort of exclusion criterion is typical in research of this kind, then it should be noted as a limitation of research in this field more generally, quite possible a very significant limitation. Certainly, taking the influence of body dissatisfaction and eating-disordered behavior into account when considering potential associations between dietary intake and mental health, rather than ignoring it, would make for a far more compelling body of research.
3) Also of concern is the apparent predominance of cross-sectional studies in this field of research. Surely there is an urgent need for more longitudinal studies in order to elucidate the direction of any observed associations between diet and mental health?
4) I would also like to see some consideration of the implications of the research for health promotion (as opposed to prevention, early intervention and treatment) programs, programs designed to improve adolescents’ awareness and understanding of the potential roles of not only diet, but also body dissatisfaction and eating-disordered behavior, in affecting depression and mental health more generally. If information concerning the potential influence of diet on the mental health of young people were to be integrated with information concerning the effects of body dissatisfaction and eating-disordered behavior then programs seeking to improve adolescents “mental health literacy” would be all the more valuable in my view (see https://pubmed.ncbi.nlm.nih.gov/32041463/).
Author Response

(The authors gave the same response as above.)
